# Effect of bed height on laryngoscopy force and operator ergonomics during simulated endotracheal intubation: A randomized controlled study

Ja Eun Lee[1]☯, Kwan Young Hong[1,2]☯, Chisong Chung[1], Jeong-Jin Min☉[1]*

**1** Department of Anesthesiology and Pain Medicine, Samsung Medical Center, Sungkyunkwan University School of Medicine, Seoul, Republic of Korea, **2** Department of Anesthesiology and Pain Medicine, International St. Mary's Hospital, Catholic Kwandong University College of Medicine, Incheon, Republic of Korea

☯ These authors contributed equally to this work.
* mjj177@g.skku.edu

## Abstract

### Purpose

Excessive force during laryngoscopy for endotracheal intubation can result in injury to airway soft tissues and hemodynamic stress responses. In this randomized controlled trial on simulated intubation, we aimed to evaluate the effect of bed height on laryngoscopy force and operator ergonomics. This study was registered on Clinical Research Information Service (CRIS) registry (KCT0006948).

### Methods

Fifty operators with varying levels of experience were enrolled to intubate an airway mannequin at two different bed heights— anterior superior iliac spine (level A) and xyphoid process (level X) of each operator—in a randomized sequence. The laryngoscopy force measured with a Pliance® pressure sensor attached to the surface of the Macintosh laryngoscopy blade, intubation characteristics, and ergonomic score based on the Rapid Entire Body Assessment tool were compared between the two bed heights (level A vs. X).

### Results

Peak and impulse laryngoscopy forces were significantly lower at xyphoid (level X) compared to the lower bed height (level A) (peak force: $36.06 \pm 9.77$ N vs. $33.74 \pm 8.69$ N, $P = 0.049$; impulse force: $251.82 \pm 106.06$ N vs. $224.18 \pm 86.48$, $P = 0.005$). Laryngeal view (Cormack-Lehane grade) and subjective comfort were also better at level X ($P = 0.0024$ and $P < 0.001$, respectively). The ergonomic score was higher at the lower bed height (level A, $P < 0.001$), indicating a more strenuous work posture.

**Data availability statement:** All relevant data are within the paper and its Supporting Information files.

**Funding:** The author(s) received no specific funding for this work.

**Competing interests:** The authors have declared that no competing interests exist.

## Conclusion

Bed height at xyphoid level reduced laryngoscopy force while improving laryngeal view and ergonomic comfort compared to ASIS level. Adjusting the bed height before endotracheal intubation can improve the operating environment, which in turn may contribute to safety of both patient and operator.

## Introduction

The incidence of mechanical airway injury related to endotracheal intubation ranges from 0.5 to 7% [1,2], indicating the need for strategies to minimize procedural trauma. Direct laryngoscopy is the most commonly used technique for endotracheal intubation. During this procedure, significant pressure is exerted on pharyngo-laryngeal tissue and can result in hemodynamic stress response, rise in intracranial pressure, and airway injury [3–6]. These may be related to the extent of pressure or force transmitted to the tissue from laryngoscopy blade [7–9]. Existing evidences suggest that the force required for laryngeal opening by direct laryngoscopy ranges from 20 to 40 Newtons (N), which is substantial considering it is comparable to holding 2–4 liters of water. Moreover, this can escalate to 50–90 N in difficult intubations [10–13].

Laryngoscopy force depends on various conditions like patient anatomy, expertise of operator, and type of laryngoscopy, most of which are not readily modifiable [11,12,14–17]. Patient bed height is an important condition for intubation that affects laryngeal view, working posture, and time to successful intubation [18,19]. An improved laryngeal view achieved by adjusting bed height may, in turn, reduce the force and time for laryngoscopy, thereby suggesting that bed height is potentially a simple, modifiable factor on laryngoscopy force. In regard of improvement in laryngeal view grade at higher bed height, i.e., when the bed height is at the level of operator's nipple rather than umbilicus, tall operators who perform intubation without raising the bed accordingly may encounter inadequate laryngeal views [19].

In this study, we compared laryngoscopy forces at bed heights aligned to two anatomical points of operators. We hypothesized that laryngoscopy force will be lower when bed height is set to xyphoid process (the higher and the optimal level) compared to anterior superior iliac spine (ASIS) (the lower and the suboptimal level).

## Methods

### Study population and design

This randomized controlled study was conducted under the approval of the Institutional Review Board (IRB) of Samsung Medical Center, Seoul, Republic of Korea (file number: 2021-09-1-41) and registered on Clinical Research Information Service (CRIS) registry (https://cris.nih.go.kr/; KCT0006948) on 2022-01-20. The recruitment period started on 2022-01-27 and ended on 2022-03-19. The study was performed in accordance with the Declaration of Helsinki and the International Conference on Harmonisation of Good Clinical Practice guidelines. We included operators with various intubation experiences (medical students, residents of the department of

anesthesiology and pain medicine, and board-certified anesthesiologists) and excluded those who refused to participate and those who had acute and/or chronic pain disease at joints of upper extremities. Written informed consent was obtained from all participants.

The trial was designed as 2 by 2 cross-over trial (Fig 1). We tested the following two bed heights: ASIS (level A) and xyphoid process (level X) of operators. Participants intubated at two heights in either of the two sequences: AX sequence (1st intubation at level A, 2nd intubation at level X) and XA sequence (1st intubation at level X, 2nd intubation at level A), to which they were allocated in random to minimize potential sequence effect. Washout period of more than 48 hours between each intubation was introduced to minimize carryover effect.

## Outcome measures

Effect of bed height was analyzed in three domains: laryngoscopy force (peak, average, and impulse forces [Ns]), intubation characteristics (attempt number, duration [sec], first-attempt success rate, laryngeal view grade, subjective discomfort

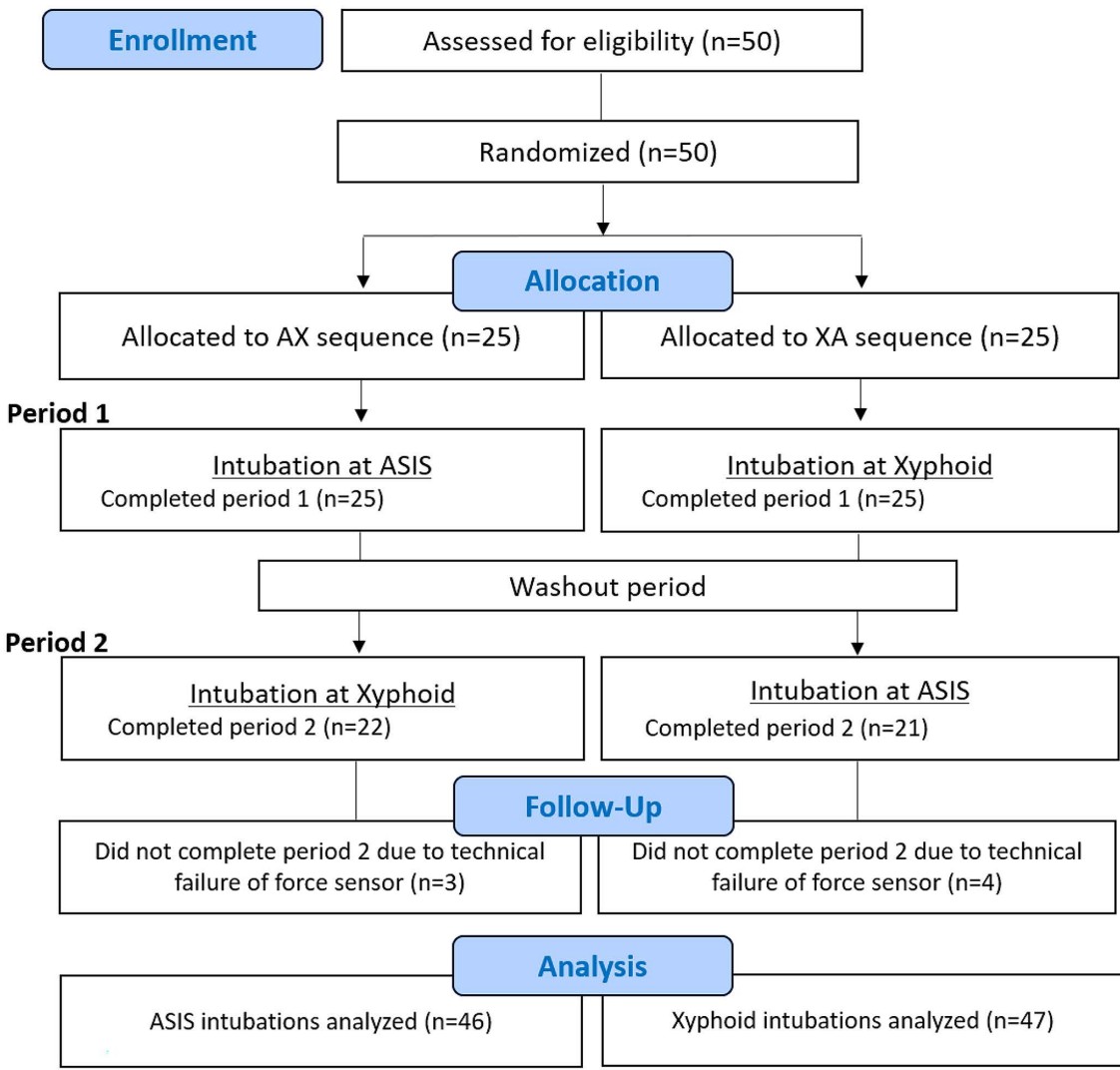

**Fig 1. CONSORT flow chart of the study.** ASIS, anterior superior iliac spine.

during intubation), and ergonomics (body posture) during intubation. Peak, average, and impulse force were defined as the maximal force at any moment during the procedure, the average force throughout the procedure, and the integral of individual force and time throughout the procedure, respectively. Laryngeal view using Cormack-Lehane laryngeal view grading [20] and subjective intubation discomfort grade (1 = no discomfort/ 2 = slight discomfort/ 3 = moderate discomfort/ 4 = significant discomfort) were reported by operators after each intubation. Ergonomic score was calculated from the Rapid Entire Body Assessment (REBA) tool. Primary outcome was the peak laryngoscopy force during intubation procedure, and secondary outcomes were the average and impulse forces, intubation characteristics, and ergonomic score.

### Sample size

We defined significant change in peak force as 20% rise in force at suboptimal bed height compared to that at optimal bed height. Linear mixed model with reference force data (mean peak force 48.8N (SD 15.8)) from a study conducted with Pliance® pressure sensor—the same sensor as ours—was used to calculate sample size [11]. With level of significance 0.05 and 80% power, sample size was calculated as $n = 43$. Assuming drop-out rate of 10%, 48 participants at minimum was targeted.

### Randomization

Participants were randomized to either AX or XA sequence using computer-generated random numbers. Stratified randomization was done based on affiliation of operators (medical student, resident in anesthesiology, or board-certified anesthesiologist) to balance the level of experience between two groups.

### Intubation procedure

For level A, bed height was adjusted so that mannequin's mid-forehead was at the level of operator's ASIS. For level X, it was adjusted to the level of xyphoid process. Intubations were done on the Koken Airway Management Model LM-059 (Koken Co., Ltd., Tokyo, Japan) with Macintosh 3 HEINE classic+ conventional laryngoscopy (HEINE Optotechnik, Gilching, Germany). Operators were free to adjust the head and neck position of mannequin according to their needs, such as taking the sniffing position. Intubation time was defined as the time from lifting the laryngoscopy to passing of endotracheal tube through glottic opening. After intubation and cuff ballooning, successful endotracheal intubation was confirmed by lung inflation.

### Laryngoscopy force measurement

Pliance® pressure sensor (Novel Electonics Incorporated, Saint Paul, MN) was applied to the contact surface of Macintosh 3 blade as in S1A Fig. This sensor consisted of 0.7 mm thick sensory array enclosed in biocompatible high-strength polyurethane film, custom-made to fit the shape of Macintosh 3 blade. Sensor array was composed of 42 independent pressure-sensing elements. Laryngoscopy force was the sum of pressure measured by each sensing element multiplied by the area of each sensing element. Sensor was connected to software for simultaneous data capture of 50 Hz frequency and real-time display of pressure and force with visual assistance (S1B Fig).

### Ergonomic score measurement

Ergonomic score during intubation was measured with the REBA, one of the most commonly used tools for assessing the risk of work-related musculoskeletal disorders [21]. It consists of central body (trunk, neck, leg) posture score, arm (upper arm, lower arm, wrist) posture score, load/force score, and activity score, from which the final REBA score is calculated. Higher REBA score implicates increased risk for development of musculoskeletal disorder (1 = negligible risk/ 2–3 = low risk, change may be needed/ 4–7 = medium risk, further investigation, change soon/ 8–10 = high risk, investigate and implement change/ > 11 = very high risk, implement change).

Each intubation was video-recorded, from which posture angles at the moment of endotracheal tube passing through glottic opening were measured to calculate the REBA score. Arm posture score was only calculated for the left arm (the laryngoscopy-lifting arm) (Fig 2).

## Statistical analysis

For comparison of baseline characteristics between AX and XA groups, Student's t-test or Mann-Whitney U test was used for continuous variables and Pearson chi-square or Fisher's exact test for categorical variables, according to normality of the data. Normality of data was tested with the Shapiro-Wilk or Kolmogorov-Smirnov test. Generalized estimating equation (GEE) was used to test the treatment effect (effect of level A vs. X). To consider the carryover, sequence, and period effects of crossover trials, they were introduced as parameters of GEE if significant. Because seven out of the fifty enrolled operators (three in AX group, four in XA group) missed their period 2 intubations due to irreparable damage to the pressure sensor during the trial, missing data incurred, which was treated with inverse probability weighting (IPW) method before GEE analysis. For secondary outcomes, a GEE model with an AR(1) working correlation structure was used without additional parameter adjustment. $P$-value $< 0.05$ was considered statistically significant for all analyses. Data were analyzed with SAS software (version 9.4, SAS institute, Cary, NC, USA) and Rex (3.7.0.0).

## Results

### Operator characteristics

Baseline characteristics of operators are described in Table 1. There were no differences in terms of age, sex, height, experience in endotracheal intubation, and affiliations between the two groups.

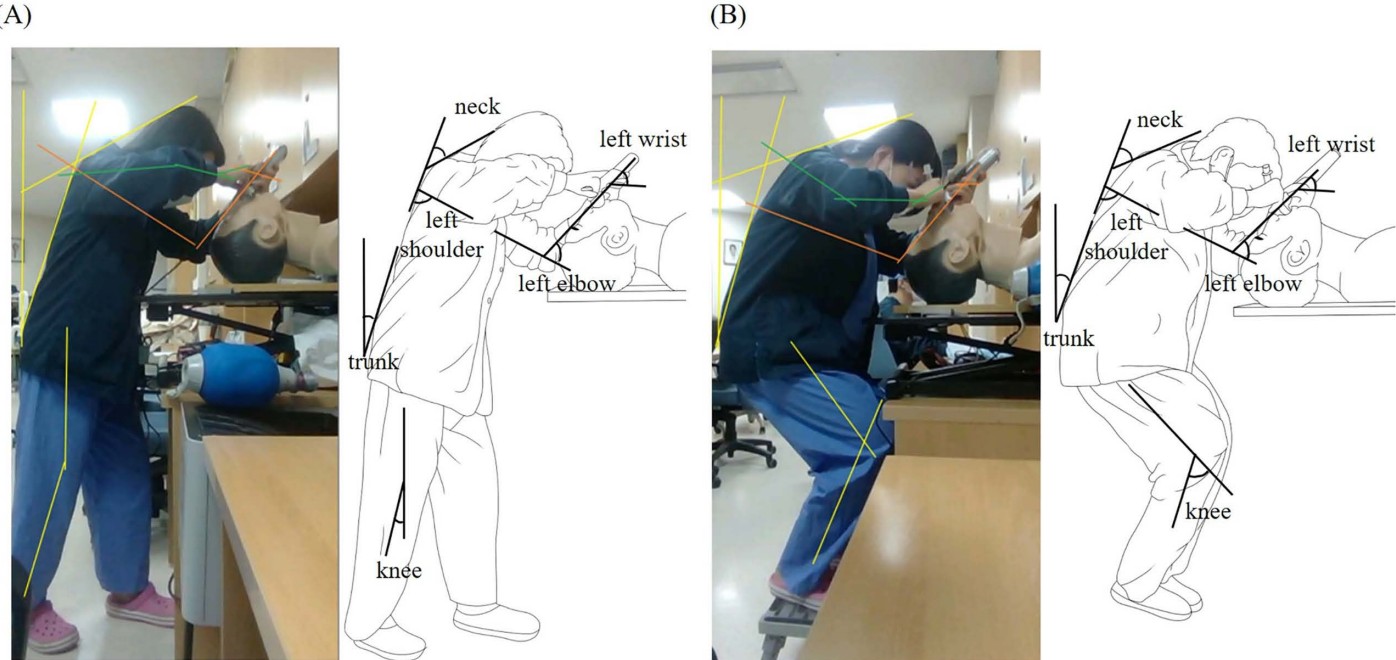

**Fig 2. Postural analysis for REBA calculation.** The figure shows the body posture of the same operator during task performance. Angles measured at (A) the xyphoid process and **(B)** ASIS were used for REBA score calculation.

**Table 1. Baseline characteristics of intubation operators.**

|  | Total (*n* = 50) | AX sequence (*n* = 25) | XA sequence (*n* = 25) |
|---|---|---|---|
| Age [year] | 31.34 ± 5.22 | 31.40 ± 4.97 | 31.28 ± 5.57 |
| Male | 22 (44.0) | 11 (44.0) | 11 (44.0) |
| Height [centimeter] | 167.02 ± 7.75 | 167.40 ± 8.03 | 166.64 ± 7.61 |
| Experience in endotracheal intubation |  |  |  |
| <1 year | 9 (18.0) | 5 (20.0) | 4 (16.0) |
| 1-4 year | 29 (58.0) | 14 (56.0) | 15 (60.0) |
| >4 year | 12 (24.0) | 6 (24.0) | 6 (24.0) |
| Affiliation |  |  |  |
| Resident in anesthesiology | 20 (40.0) | 10 (40.0) | 10 (40.0) |
| Medical student | 23 (46.0) | 11 (44.0) | 12 (48.0) |
| Medical student | 7 (14.0) | 4 (16.0) | 3 (12.0) |

Data are presented as mean ± standard deviation for continuous variables and number (percentage) for categorical variables. AX, anterior superior iliac spine – xyphoid; XA, xyphoid – anterior superior iliac spine.

## Carryover effect

There were no carryover effects on peak, average, and impulse forces, suggesting that the experience of first intubations did not influence magnitude of laryngoscopy force during the second intubations (S1 Table).

## Laryngoscopy force

The mean peak force was 36.06 N at level A and 33.74 N at level X. After adjustment for sequence and period effects, the treatment effect of bed height showed significant difference [mean difference (standard error (SE)) 2.38 (1.21) N; 95% confidence interval (CI) 0.01 to 4.76 N; *P* = 0.049] (Table 2). The average force showed no difference between two heights [mean difference (SE) 1.19 (0.65) N; 95% CI −0.08 to 2.46 N; *P* = 0.066]. The mean impulse forces were 251.82 Ns and 224.18 Ns at level A and X, respectively. Similar to peak force, impulse force was significantly higher at level A after adjusting for sequence and period effects [mean difference (SE) 37.45 (13.22) Ns; 95% CI 11.54 to 63.36 Ns; *P* = 0.0046].

**Table 2. Laryngoscopy force and intubation characteristics according to the bed height.**

|  | ASIS | Xyphoid | Estimate (SE) | 95% CI | *p* value |
|---|---|---|---|---|---|
| Peak force [N] | 36.06 ± 9.77 | 33.74 ± 8.69 | 2.38 (1.21) | 0.01, 4.76 | 0.049* |
| Average force [N] | 16.88 ± 5.23 | 15.73 ± 4.08 | 1.19 (0.65) | −0.08, 2.46 | 0.066 |
| Impulse force [N] | 251.82 ± 106.06 | 224.18 ± 86.48 | 37.45 (13.22) | 11.54, 63.36 | 0.0046[†] |
| Laryngeal view grade[a] | 2 [1–2] | 1 [1–2] | −0.33 (0.11) | −0.54, −0.11 | 0.0024[†] |
| Subjective discomfort[b] | 2 [1–3] | 1 [1–2] | −0.58 (0.14) | −0.856, −0.30 | <0.001[‡] |

Data are presented as mean ± standard deviation or median [interquartile range].

ASIS, anterior superior iliac spine; SE, standard error; CI, confidence interval.

[a]Laryngeal view is based on Cormack-Lehane grading, and our numerical scale corresponds to each grade as follows: 1 = grade 1/ 2 = grade 2A/ 3 = grade 2B/ 4 = grade 3A/ 5 = grade 3B/ 6 = grade 4.

[b]Subjective discomfort grade is as follows: 1 = no discomfort/ 2 = slight discomfort/ 3 = moderate discomfort/ 4 = significant discomfort. For laryngoscopy force, generalized estimating equation (GEE) model was used with missing data treated with inverse probability weighting (IPW); *n* = 46 and 47 for ASIS and xyphoid, respectively. For laryngeal view grade and subjective discomfort, GEE model with an AR(1) working correlation structure was used; *n* = 44 and 45 for ASIS and xyphoid, respectively.

*, † , ‡ denote *p*-values below 0.05, 0.01, 0.001, respectively.

## Sensitivity analysis for laryngoscopy force

For sensitivity analysis, we performed GEE analyses with four other datasets that differed in the treatment of missing data: (1) Missing data imputed with regression method, (2) Missing data imputed with predictive mean matching method, (3) Missing data not imputed nor weighted, (4) Complete data (excluding subjects with missing data). In GEE models based on missing data treated with imputation, impulse force was consistently higher at level A while peak force was higher at level A only in the model with imputed missing data from regression method.

## Intubation-related characteristics

All intubations were successful on the first attempt. There was no significant difference in the time to successful intubation between the two heights ($P=0.37$). However, operator's subjective discomfort level during the procedure was lower at level X [effect size (SE) −0.575 (0.14); 95% CI −0.855 to −0.296; $P<0.001$], and the laryngeal view with Cormack-Lehane grading was also significantly better at level X compared to level A [effect size (SE) −0.328 (0.11); 95% CI −0.542 to −0.113; $P=0.0024$) (Table 2).

## Ergonomics

Final REBA score, which reflects operator's ergonomic stress during intubation procedure, was significantly higher at level A than at level X [effect size, −2.21 (0.21); 95% CI −2.64 to −1.79; $P<0.001$] (Table 3). Among the components for calculating the final REBA score, central body REBA subscore (trunk, neck, and knee) was observed significantly higher at level A [effect size (SE) −1.65 (0.17); 95% CI −1.98 to −1.32; $P<0.001$], but no significant difference was found in the left arm REBA subscore between the two heights [effect size (SE) −0.17 (0.14); 95% CI −0.45 to 0.11; $P=0.23$) (Table 3).

## Discussion

In this prospective study for simulated endotracheal intubation, peak and impulse laryngoscopy forces were significantly lower when the bed height was aligned with operator's xyphoid level than with ASIS level. We also found that laryngeal view grade, operator's subjective comfort, and ergonomics were significantly improved at the xyphoid level.

### Laryngoscopy force

While all three types of forces showed lower mean values at the xyphoid level, the magnitude of difference and statistical significance differed according to the type of force. The greatest difference in impulse force suggests that the impact of bed height becomes evident in time-dependent manner. This may be explained by concurrent improvement in laryngeal view at the xyphoid level, which is similar to how video-laryngoscopy reduces force and/or pressure compared to direct laryngoscopy. Unlike direct laryngoscopy which requires tongue displacement and anterior-upward lifting of mandible to

**Table 3. Ergonomic score comparison using Rapid Entire Body Assessment (REBA) tool between two table heights.**

|  | ASIS | Xyphoid | Estimate (SE) | 95% CI | *p* value |
|---|---|---|---|---|---|
| Final REBA score | 8.32 ± 1.25 | 6.09 ± 1.47 | −2.21 (0.21) | −2.63, −1.79 | <0.001‡ |
| REBA subscore for central body[a] | 7.09 ± 1.01 | 5.44 ± 0.89 | −1.65 (0.17) | −1.98, −1.32 | <0.001‡ |
| REBA subscore for left arm[b] | 4.20 ± 0.88 | 4.02 ± 0.66 | −0.17 (0.14) | −0.45, 0.11 | 0.234 |

Data are presented as mean ± standard deviation. *ASIS* anterior superior iliac spine, *SE* standard error, *CI* confidence interval.

[a]REBA subscore for central body is based on joint angles of neck, trunk, and knees.

[b]REBA subscore for left arm (the laryngoscopy-lifting arm) is based on joint angles of left shoulder, elbow, and wrist. GEE model with an AR(1) working correlation structure was used; $n=44$ and 45 for ASIS and xyphoid, respectively.

*, † ‧ ‡ denote *P*-values below 0.05, 0.01, 0.001, respectively.

expose the vocal cords, video-laryngoscopy has camera at its tip for indirect glottic visualization and avoids severe anatomical disturbance and need for lifting [11,22]. Likewise, better laryngeal view at the xyphoid level could have reduced the force and time to secure a proper line of sight at glottic aperture, resulting in attenuation of impulse force. This also explains the improvement in subjective comfort at the xyphoid level.

Our findings align with previous studies that examined the impact of bed height on intubation performance. For example, a study by Hong et al. reported that lower bed positioning worsened procedural difficulty and laryngeal view [23]. Similarly, Nikolla et al. found that optimal bed height and ramp angle were associated with enhanced laryngeal view score and intubation time [18]. These results support our observation that adjusting bed height to the operator's xiphoid level may improve both laryngeal view and force efficiency during intubation.

### Clinical significance of laryngoscopy force

Physiological significance of laryngoscopy force is presumed to differ by the type of force. For example, peak force has been implicated for pharyngo-laryngeal soft tissue injury whereas impulse force has been associated with cardiovascular response and cervical injury [14].

Although the reduction in peak force of approximately 6.6% was statistically significant, it is modest in magnitude when compared with previous studies using advanced airway devices. For example, studies utilizing video-laryngoscopy have reported reductions in peak force of approximately 50–80% [11,14]. The force magnitude that constitutes a clinical threshold for causing tissue injury (e.g., tongue, lip, oral mucosa, or teeth) is not definitively established, but studies suggest that excessive force, particularly during difficult intubations, can lead to tissue damage [11,24]. However, these studies are based on patients with normal tissue, and small reduction may still be meaningful for sensitive patients such as those with friable oral tissue or dental instability. Thus, the observed reduction in this study may still reduce the risk of injury in sensitive patients, though it is likely less impactful than reductions seen with video-laryngoscopy.

Cardiovascular responses are triggered by glossopharyngeal and vagus nerves sensing noxious stimuli at tongue base [25]. Concentration of plasma catecholamine increases with the initiation of laryngoscopy and reaches its peak after one minute, leading to exaggerated rise in blood pressure and heart rate in response to prolonged intubation [26–28]. Therefore, decreasing impulse force could potentially benefit patients with cardiovascular disease, intracranial pathology, or hypertension, although further studies are needed to confirm this effect.

### Ergonomic interpretation

The lower REBA score at the xyphoid level indicates better ergonomic environment than at the ASIS level. REBA score assesses the risk of musculoskeletal injury for a specific body posture on a scale of 1–15, and higher score indicates greater risk of injury and need for amendment. It is derived from joint angles, amount of force to be withstood, duration, and repetitiveness. Joint angles are again divided into central (neck, trunk, knee) and peripheral (shoulder, elbow, wrist) types. In our study, the central body subscore but not the left arm subscore (the laryngoscopy-lifting arm) was higher at the ASIS level, suggesting that our study subjects compensated for the discrepancy between the bed height and their line of sight by angulating their central body. For instance, if operators felt the ASIS level was too low, they tended to optimize the situation by bending their neck, back, and knees to move towards the airway rather than lifting their arm to bring the airway closer to them. This can also explain why the peak force had difference of borderline significance ($P = 0.049$), as it was the maximal force exerted by the left arm while the burden was primarily tolerated by the central body. Therefore, suboptimal bed height was overcome by joint sacrifices from both the airway mannequin (greater trauma) and the operator (strenuous work posture). Based on these findings, adjusting the bed height to the operator's xiphoid level is recommended, particularly for prolonged procedures or taller operators, as it may decrease cumulative musculoskeletal strain. By reducing ergonomic strain, adjusting bed height could help prevent chronic musculoskeletal issues, including neck,

shoulder, and back pain, which often develop over years of repetitive procedures. Such preventive measures may promote clinician well-being and sustainability in high-demand clinical environments.

### Strength of our study

Factors on laryngoscopy force are divided into patient, operator, and environmental factors, among which only the environmental factors can accommodate immediate changes. However, to improve the environment for intubation usually requires alternative intubation devices or supplementary aiding instruments. The novelty of this study lies in simplicity and generalizability of its intervention, as bed height adjustment can be applied to almost any clinical circumstance, ranging from induction of general anesthesia to emergency intubation for life-threatening situations [23]. Second, our study contributes to the currently limited body of literature for intubation ergonomics. We systematically analyzed the risk of musculoskeletal injury associated with intubation based on an ergonomic scoring system and showed that adjusting bed height can reduce an occupational risk that anesthesiologists have to persistently sustain.

### Limitation

There are several limitations to the present study. Our results are based on simulated endotracheal intubation performed on mannequin, and whether optimization of bed height and laryngoscopy force can clinically manifest in humans as attenuated cardiovascular responses and reduced airway injuries remains unclear. Mannequin airways differ from human airways in several critical aspects, including the absence of tissue compliance, mucosal sensitivity, and hemodynamic responses such as cardiovascular reactions to laryngoscopy. These differences may have influenced the force dynamics observed in our study. Although this study provides controlled experimental data using a mannequin, the results may not fully reflect real clinical conditions where patient anatomy, operator expertise, and environmental factors vary significantly. Therefore, further clinical studies across diverse patient populations and real-world settings—such as measuring plasma catecholamine responses or airway injury rates—are necessary to confirm whether the observed outcomes can be generalized. Second, our mannequin was not a difficult airway model and our results cannot be applied to such situations. However, the effect of bed height is likely more apparent in difficult airways because they require higher force for successful intubation [13,29]. Additionally, 7 operators in period 2 were excluded due to sensor damage, which led to missing data. Although inverse probability weighting (IPW) was applied to minimize potential bias, this reduction in sample size may have decreased the statistical power to detect subtle differences between groups. To address this concern, we conducted sensitivity analyses comparing the missing data imputation (with regression method) and IPW-imputed datasets, and the results were consistent with our primary findings, suggesting that our conclusions remain robust despite the missing data. Nevertheless, the potential impact of the reduced sample size on the precision of estimates should be acknowledged.

### Conclusion

In this study, aligning the bed height with the operator's xyphoid level, compared to the ASIS level, significantly reduced both peak and impulse laryngoscopy forces and improved the laryngeal view, operator comfort, and ergonomics during simulated endotracheal intubation. Implementing this simple adjustment may improve procedural safety and operator performance, although confirmation in diverse clinical settings remains essential.

### Supporting information

**S1 Fig. Instrumentation for laryngoscopy force measurement.** (A) Pliance® pressure sensor applied to the contact surface of Macintosh 3 blade. (B) Force data capture process and real-time display of pressure and force with visual assistance.
(TIF)

**S1 Table. Carryover, sequence, and period effects.** *SE* standard error, *CI* confidence interval. [a]Carryover effect evaluated from GEE model with missing data not treated with imputation nor weighting Carryover effect was not found in other datasets used in sensitivity analysis.
(DOCX)

**S1 File. Raw data of posture angle.**
(XLSX)

**S2 File. Raw data of laryngoscopy force.**
(XLSX)

## Author contributions

**Conceptualization:** Ja Eun Lee, Kwan Young Hong, Jeong-Jin Min.

**Data curation:** Kwan Young Hong, Chisong Chung.

**Formal analysis:** Ja Eun Lee, Kwan Young Hong, Jeong-Jin Min.

**Investigation:** Ja Eun Lee, Kwan Young Hong, Jeong-Jin Min.

**Methodology:** Jeong-Jin Min.

**Project administration:** Jeong-Jin Min.

**Resources:** Jeong-Jin Min.

**Writing – original draft:** Ja Eun Lee, Chisong Chung.

**Writing – review & editing:** Jeong-Jin Min.

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
