## [Decision Letter · Decision Letter 0]

14 Jul 2025

Dear Dr. Min,

Thank you for submitting your manuscript to PLOS ONE. After careful consideration, we feel that it has merit but does not fully meet PLOS ONE’s publication criteria as it currently stands. Therefore, we invite you to submit a revised version of the manuscript that addresses the points raised during the review process.

**ACADEMIC EDITOR:**

We look forward to receiving your revised manuscript.

Kind regards,

Ahmet Çağlar, Associate Professor

Academic Editor

PLOS ONE

Journal Requirements: 

2. In the online submission form, you indicated that [The data for the present study is available from the corresponding author on reasonable request.].

Additional Editor Comments:

I read the manuscript. I think it is well designed and written. I will reconsider it once major revisions have been made.

Reviewers' comments:

Reviewer's Responses to Questions

**Comments to the Author**

1. Is the manuscript technically sound, and do the data support the conclusions?

Reviewer #1: Partly

Reviewer #2: Partly

2. Has the statistical analysis been performed appropriately and rigorously?

Reviewer #1: Yes

Reviewer #2: I Don't Know

3. Have the authors made all data underlying the findings in their manuscript fully available?

Reviewer #1: Yes

Reviewer #2: Yes

4. Is the manuscript presented in an intelligible fashion and written in standard English?

Reviewer #1: Yes

Reviewer #2: Yes

Reviewer #1: Thank you for giving me the opportunity to review this study. This study was a randomized crossover design conducted on a manikin to compare the force applied to tissues by laryngoscopy, intubation characteristics, and ergonomics at two different bed heights. The hypothesis was that less force would be required when the bed height was at the xiphoid process level compared to the ASIS level. The primary outcome was the peak laryngoscopy force during the intubation process. The results showed that the mean peak force and impulse force were greater at the ASIS height compared to the xiphoid height, but there was no difference in average force. At the xiphoid height, subjective discomfort levels and ergonomic stress were lower, and the laryngeal view was better.

The study was well-designed, conducted, and analyzed, and the manuscript was logically written.

I have a few minor comments:

• It is necessary to interpret this study in relation to previous studies that compared intubation procedures at various bed heights, including "Signae Vitae Vol.12, Issue S1, October 2016 pp.47-51" and "J Am Coll Emerg Physicians Open. 2020 Mar 13;1(3):257-262."

• The sentence on Line 312, " Therefore, decreasing impulse force would be helpful for patients with cardiovascular disease, intracranial pathology, or hypertension." needs to be toned down slightly.

• While the limitations of using a manikin in a study, rather than actual clinical settings, were acknowledged, it is necessary to elaborate further on the need to confirm whether the same results would be observed in diverse real clinical environments, considering all aspects such as patients, operators, and environments.

• The conclusion contains statements unrelated to the observations made in this study. The conclusion needs to be rewritten based on the results obtained from this research.

Reviewer #2: I appreciate your permission to examine the manuscript. I appreciate your confidence in my expertise and have enjoyed participating in the peer review process of this journal. This study provides a practical solution to reduce laryngoscopy force and improve ergonomics. It will be more effective if the suggested revisions are addressed, particularly the clinical context for force reductions and simulation limitations.

1. In limitations of simulation (p. 16), please clarify key differences between manikin vs. human airways (e.g., tissue compliance, hemodynamic responses). Future clinical validation (e.g., measuring plasma catecholamines or airway injury rates) is recommended.

2. How to handle missing data (pp. 9–10), 7 operators were excluded in period 2 due to sensor damage. Though IPW was used, please include sensitivity analyses of datasets (e.g., complete-case vs. imputed) in the supplementary material. Discuss the potential impact of missing data on statistical power in the limitations.

3. In the clinical relevance of peak force reduction (p. 11) section, the peak force reduction (2.38 N2.38N, ~7%) was borderline significant (p=0.049p=0.049). Please discuss clinical thresholds for tissue injury (e.g., compare with forces in studies of difficult airways or video laryngoscopy, such as Hindman et al.). Contrast the 7% reduction with larger reductions from advanced devices (e.g., ~30–40% with video laryngoscopy).

4. In the ergonomic interpretation section (pp. 14–15), REBA scores indicated "high risk" at height A (8.32) vs. "medium risk" at X (6.09). You can add practical guidance: "Adjust bed height to xiphoid level, especially for prolonged procedures or taller operators." Please emphasize the long-term impact of reducing work-related musculoskeletal disorders.

5. In the Introduction (p. 5), cite prevalence data for laryngeal injury early (e.g., "0.5–7%," Refs. 5–6) to underscore clinical urgency. Explicitly link bed height → laryngeal view → force reduction in the proposed mechanism.

6. In the supplementary materials section (pp. 27–30), you can add operator posture images at heights A/X to visualize REBA differences (replace placeholder Fig 2a/b). Share raw force datasets per PLOS ONE policy (e.g., via repository or supplementary files).

**Do you want your identity to be public for this peer review?** For information about this choice, including consent withdrawal, please see our Privacy Policy

Reviewer #1: No

Reviewer #2: No

---

## [Author Response · Author response to Decision Letter 1]

24 Aug 2025

Dear Editorial office,

Thank you for the opportunity to revise our manuscript entitled “Effect of bed height on laryngoscopy force and operator ergonomics during simulated endotracheal intubation: a randomized controlled study”. We appreciate all the constructive comments from the editor and reviewers. Our point-by-point responses to the reviewers’ comments are listed below, and we have highlighted the revisions in the manuscript using yellow highlights. We believe that these changes have significantly improved the manuscript and hope it is now deemed suitable for publication in Plos One.

Sincerely,

Jeong-Jin Min, M.D., Ph.D., Associate Professor

Department of Anesthesiology and Pain Medicine, Samsung Medical Center

Sungkyunkwan University School of Medicine

81 Irwon-Ro, Gangnam-Gu, Seoul 06351, Republic of Korea

Tel.: +82 2 3410 2471

Fax: +82 2 3410 0361

E-mail: mjj177@g.skku.edu 

Reviewer 1

Reviewer’s Comments

1. It is necessary to interpret this study in relation to previous studies that compared intubation procedures at various bed heights, including "Signae Vitae Vol.12, Issue S1, October 2016 pp.47-51" and "J Am Coll Emerg Physicians Open. 2020 Mar 13;1(3):257-262."

→ Response: Thank you for your valuable comment. We have revised the discussion section (‘Laryngoscopy force’ section, p. 14) to compare our findings with previous studies that evaluated intubation performance at different bed heights. The following text has been added:

--“Our findings align with previous studies that examined the impact of bed height on intubation performance. For example, a study by Hong et al. reported that lower bed positioning worsened procedural difficulty and laryngeal view [23]. Similarly, Nikolla et al. found that optimal bed height and ramp angle were associated with enhanced laryngeal view score and intubation time [18]. These results support our observation that adjusting bed height to the operator’s xiphoid level may improve both laryngeal view and force efficiency during intubation.”

--Also, a sentence in Discussion (‘Strength of our study’ section, p.16) was expanded:

“The novelty of this study lies in simplicity and generalizability of its intervention, as bed height adjustment can be applied to almost any clinical circumstance, ranging from induction of general anesthesia to emergency intubation for life-threatening situations [23]”

2. The sentence on Line 312, " Therefore, decreasing impulse force would be helpful for patients with cardiovascular disease, intracranial pathology, or hypertension." needs to be toned down slightly.

→ Response: Thank you for your thoughtful suggestion. We agree that the original sentence may have sounded overly conclusive. We have revised the sentence (p.15, line 332) to read:

“Therefore, decreasing impulse force could potentially benefit patients with cardiovascular disease, intracranial pathology, or hypertension, although further studies are needed to confirm this effect.”

This change was intended to reflect a more cautious interpretation and acknowledge the need for further validation.

3. While the limitations of using a manikin in a study, rather than actual clinical settings, were acknowledged, it is necessary to elaborate further on the need to confirm whether the same results would be observed in diverse real clinical environments, considering all aspects such as patients, operators, and environments.

→ Response: Thank you for your valuable comment. We agree that further clarification regarding the applicability of our findings to real clinical environments is needed. In response, we have expanded the limitations section to emphasize that our results obtained from manikin experiments may not fully replicate clinical conditions. We have also highlighted the necessity of future studies in diverse patient populations and clinical scenarios to validate these findings. The added sentences (‘Limitation’ section, lines 380--385) are as follows:

“Although this study provides controlled experimental data using a mannequin, the results may not fully reflect real clinical conditions where patient anatomy, operator expertise, and environmental factors vary significantly. Therefore, further clinical studies across diverse patient populations and real-world settings are necessary to confirm whether the observed outcomes can be generalized.”

4. The conclusion contains statements unrelated to the observations made in this study. The conclusion needs to be rewritten based on the results obtained from this research.

→ Response: We appreciate your thoughtful feedback and agree that the original conclusion was not fully aligned with our study findings. To address your comment, we have revised the conclusion to focus more directly on our results and their implications. The updated conclusion is:

“In this study, aligning the bed height with the operator’s xyphoid level, compared to the ASIS level, significantly reduced both peak and impulse laryngoscopy forces and improved the laryngeal view, operator comfort, and ergonomics during simulated endotracheal intubation. Implementing this simple adjustment may improve procedural safety and operator performance, although confirmation in diverse clinical settings remains essential.”

We believe this revision more accurately summarizes our observations and responds to your concern.

Reviewer 2

Reviewer’s Comments

1. In limitations of simulation (p. 16), please clarify key differences between manikin vs. human airways (e.g., tissue compliance, hemodynamic responses). Future clinical validation (e.g., measuring plasma catecholamines or airway injury rates) is recommended.

→ Response: Thank you for this insightful suggestion. We have revised the Limitations section to explicitly describe the differences between manikin and human airways, including the absence of tissue compliance, mucosal sensitivity, and hemodynamic responses such as cardiovascular reactions to laryngoscopy. We have also added a statement emphasizing the need for future clinical studies, including measurements of plasma catecholamine responses and airway injury rates, to validate the applicability of our findings. The revised text now reads (p.17, line 376-385):

“Mannequin airways differ from human airways in several critical aspects, including the absence of tissue compliance, mucosal sensitivity, and hemodynamic responses such as cardiovascular reactions to laryngoscopy. These differences may have influenced the force dynamics observed in our study. … Therefore, further clinical studies across diverse patient populations and real-world settings—such as measuring plasma catecholamine responses or airway injury rates—are necessary to confirm whether the observed outcomes can be generalized.”

2. How to handle missing data (pp. 9–10), 7 operators were excluded in period 2 due to sensor damage. Though IPW was used, please include sensitivity analyses of datasets (e.g., complete-case vs. imputed) in the supplementary material. Discuss the potential impact of missing data on statistical power in the limitations

→ Response: Thank you for this valuable comment. We have added sensitivity analyses comparing the complete-case dataset with the IPW-imputed dataset, and the findings remained consistent with our primary results. We also expanded the limitations section to discuss the potential impact of missing data on statistical power. Specifically, we noted that the exclusion of 7 operators in period 2 reduced the effective sample size, which may have decreased the precision of our estimates. The added text in ‘Limitations’ section reads (p. 17, line 388):

“Additionally, 7 operators in period 2 were excluded due to sensor damage, which led to missing data. Although inverse probability weighting (IPW) was applied to minimize potential bias, this reduction in sample size may have decreased the statistical power to detect subtle differences between groups. To address this concern, we conducted sensitivity analyses comparing the missing data imputation (with regression method) and IPW-imputed datasets, and the results were consistent with our primary findings, suggesting that our conclusions remain robust despite the missing data. Nevertheless, the potential impact of the reduced sample size on the precision of estimates should be acknowledged.”

3. In the clinical relevance of peak force reduction (p. 11) section, the peak force reduction (2.38 N2.38N, ~7%) was borderline significant (p=0.049p=0.049). Please discuss clinical thresholds for tissue injury (e.g., compare with forces in studies of difficult airways or video laryngoscopy, such as Hindman et al.). Contrast the 7% reduction with larger reductions from advanced devices (e.g., ~30–40% with video laryngoscopy).

→ Response: We thank the reviewer for their insightful comment regarding the clinical relevance of the observed peak force reduction. In response, we have revised the manuscript (page 14-15) to explicitly acknowledge that the 6.6% reduction in peak force observed in our study, although statistically significant (p = 0.049), is modest in comparison to reductions reported with advanced airway devices such as video laryngoscopes. Specifically, we now reference studies demonstrating reductions of approximately 50–80% in peak force with devices such as the GlideScope and Airtraq. Additionally, we discuss the lack of a clearly defined clinical threshold for soft tissue injury but note that excessive force during intubation—especially in difficult airways—has been suggested to be associated with tissue damage. Importantly, we highlight that even small reductions in force may be clinically relevant in vulnerable populations (e.g., patients with friable oral tissues or dental instability), and have included this consideration in the revised text (p. 14-15, line 318-328).

4. In the ergonomic interpretation section (pp. 14–15), REBA scores indicated "high risk" at height A (8.32) vs.

"medium risk" at X (6.09). You can add practical guidance: "Adjust bed height to xiphoid level, especially for prolonged procedures or taller operators." Please emphasize the long-term impact of reducing work-related musculoskeletal disorders.

→ Response: Thank you for your valuable suggestion. We have added practical guidance recommending bed height adjustment to the operator’s xiphoid level, particularly for prolonged procedures or taller operators. We also expanded the discussion (‘Ergonomic interpretation’ section, p 16, line 351) to emphasize the long-term impact of reducing ergonomic strain. The added text now reads:

“Based on these findings, adjusting the bed height to the operator’s xiphoid level is recommended, particularly for prolonged procedures or taller operators, as it may decrease cumulative musculoskeletal strain. By reducing ergonomic strain, adjusting bed height could help prevent chronic musculoskeletal issues, including neck, shoulder, and back pain, which often develop over years of repetitive procedures. Such preventive measures may promote clinician well-being and sustainability in high-demand clinical environments.”

5. In the Introduction (p. 5), cite prevalence data for laryngeal injury early (e.g., "0.5–7%," Refs. 5–6) to underscore clinical urgency. Explicitly link bed height → laryngeal view → force reduction in the proposed mechanism

→ Response: Thank you for your comment. We have revised the introduction to emphasize the clinical urgency at the very beginning by citing the prevalence of laryngeal injury (0.5–7%, Refs. 5–6). We have also made the proposed mechanism—linking bed height, laryngeal view, and force reduction—more explicit.

--The revised first paragraph of the Introduction now begins:

“The incidence of mechanical airway injury related to endotracheal intubation ranges from 0.5 to 7% [5,6], indicating the need for strategies to minimize procedural trauma. Direct laryngoscopy is the most commonly used technique for endotracheal intubation. During this procedure, significant pressure is exerted on pharyngo-laryngeal tissue and can result in hemodynamic stress response, rise in intracranial pressure, and airway injury [1-4].”

--The revised second paragraph of the Introduction now includes the following sentence:

“An improved laryngeal view achieved by adjusting bed height may, in turn, reduce the force and time for laryngoscopy, thereby suggesting that bed height is potentially a simple, modifiable factor on laryngoscopy force.”

6. In the supplementary materials section (pp. 27–30), you can add operator posture images at heights A/X to

visualize REBA differences (replace placeholder Fig 2a/b). Share raw force datasets per PLOS ONE policy (e.g., via repository or supplementary files).

→ Response: We appreciate the reviewer’s constructive suggestions regarding the figures. In response:

1. Operator Posture Images: We have replaced the placeholder figures (Fig. 2a/b) with actual and illustrated images of the operator’s posture at both height A and height X. These images are now included in the Figure 2 A/B to help visualize differences in REBA scores. The placeholder figures have been moved to supplementary materials.

2. Raw Force Data Sharing: In accordance with PLOS ONE’s data availability policy, we have uploaded the raw force measurement datasets as supplementary information.

We thank the reviewer again for this helpful recommendation, which has improved the transparency and clarity of the study.

---

## [Decision Letter · Decision Letter 1]

10 Sep 2025

Effect of bed height on laryngoscopy force and operator ergonomics during simulated endotracheal intubation: a randomized controlled study

PONE-D-25-28136R1

Dear Dr. Min,

We’re pleased to inform you that your manuscript has been judged scientifically suitable for publication and will be formally accepted for publication once it meets all outstanding technical requirements.

Kind regards,

Ahmet Çağlar, Associate Professor

Academic Editor

PLOS ONE

Additional Editor Comments (optional):

Dear Authors,

Thanks for the revisions. After major revisions, it is now suitable to be published in Plos One. Well done.

Your sincerely.

Reviewers' comments:

Reviewer's Responses to Questions

**Comments to the Author**

Reviewer #1: All comments have been addressed

Reviewer #2: All comments have been addressed

2. Is the manuscript technically sound, and do the data support the conclusions?

Reviewer #1: Yes

Reviewer #2: Yes

3. Has the statistical analysis been performed appropriately and rigorously?

Reviewer #1: Yes

Reviewer #2: Yes

4. Have the authors made all data underlying the findings in their manuscript fully available?

Reviewer #1: Yes

Reviewer #2: Yes

5. Is the manuscript presented in an intelligible fashion and written in standard English?

Reviewer #1: Yes

Reviewer #2: Yes

Reviewer #1: Thank you for the revised manuscript. The authors have effectively addressed all of my comments and concerns. I feel the paper is now in good shape and recommend accepting it for publication.

Reviewer #2: Dear,

As the reviewer of this manuscript, I am pleased to confirm that the authors have thoroughly and thoughtfully addressed all the points and concerns I raised in my initial assessment. The manuscript is now markedly improved, with a more robust methodology, a more nuanced interpretation of the results, and a more straightforward presentation of its contribution to the field. It now meets the high standards required for publication.

Best regards

**Do you want your identity to be public for this peer review?** For information about this choice, including consent withdrawal, please see our Privacy Policy

Reviewer #1: No

Reviewer #2: **Yes: ** Afshin Khazaei

---

## [Editor Report · Acceptance letter]

PONE-D-25-28136R1

PLOS ONE

Dear Dr. Min,

I'm pleased to inform you that your manuscript has been deemed suitable for publication in PLOS ONE. Congratulations! Your manuscript is now being handed over to our production team.

Kind regards,

on behalf of

Dr. Ahmet Çağlar

Academic Editor

PLOS ONE